# Integrated Building Energy Simulation–Life Cycle Assessment (BES–LCA) Approach for Environmental Assessment of Agricultural Building: A Review and Application to Greenhouse Heating Systems

**Cristina Decano-Valentin [1,2]**, **In-Bok Lee [1,\*]**, **Uk-Hyeon Yeo [3]**, **Sang-Yeon Lee [1]**, **Jun-Gyu Kim [1]**, **Se-Jun Park [1]**, **Young-Bae Choi [1]**, **Jeong-Hwa Cho [1]** and **Hyo-Hyeog Jeong [1]**

[1] Department of Rural Systems Engineering, Research Institute for Agriculture and Life Sciences, College of Agriculture and Life Sciences, Seoul National University, Gwanakno 1, Gwanakgu, Seoul 08826, Korea; cristinadecano1@gmail.com (C.D.-V.); tkddus613@snu.ac.kr (S.-Y.L.); kjkkjk9410@snu.ac.kr (J.-G.K.); sjpark0901@snu.ac.kr (S.-J.P.); cyb1357@snu.ac.kr (Y.-B.C.); jh.cho@snu.ac.kr (J.-H.C.); wjdgygur@snu.ac.kr (H.-H.J.)

[2] Department of Agricultural and Biosystems Engineering, College of Engineering, Mariano Marcos State University, Batac City 2906, Ilocos Norte, Philippines

[3] Research Institute for Agriculture and Life Sciences, College of Agriculture and Life Sciences, Seoul National University, Gwanakno 1, Gwanakgu, Seoul 08826, Korea; uhyeo@snu.ac.kr

\* Correspondence: iblee@snu.ac.kr

**Abstract:** A substantial reduction in the environmental impacts related to the construction and operation of agricultural buildings is needed to adapt to the continuing development of agriculture. The life cycle assessment (LCA) is a methodology used to quantify the environmental impact of different processes involved in the production and therefore has been increasingly applied to assess the environmental burden. However, most LCA-related research studies have focused on the overall environmental impact of the entire system without considering the energy load of the agricultural buildings. By integrating the LCA tool with other design tools such as the building energy simulation (BES), the identification of environmental hotspots and the mitigation options become possible during the design process. Thus, the objective of the paper was to identify the current integration approaches used to combine BES and LCA results to assess the environmental impact of different heating systems such as absorption heat pump (AHP) using energy from thermal effluent, electricity-powered heat pump and kerosene-powered boilers used in a conventional multi-span Korean greenhouse. The assessment result revealed that the environmental impact caused using a kerosene-powered boiler is largest in terms of the acidification potential (AP), global warming potential (GWP) and Eutrophication Potential (EP) of $1.15 \times 10^0$ kg $SO_2$-eq, $1.13 \times 10^2$ kg $CO_2$-eq and $1.62 \times 10^{-1}$ kg $PO_4$-eq, respectively. Detailed analysis of the result showed that the main contributor for greenhouse gas emission was caused by the type, amount and source of energy used to heat the greenhouse, which contributed to a maximum of 86.59% for case 1, 96.69% for case 2 and a maximum of 96.47% for case 3, depending on the type of greenhouse gas being considered.

**Keywords:** building energy simulation; heating systems; life cycle assessment; multi-span greenhouse; heat pump; boiler

## 1. Introduction

Greenhouse production is now becoming the major crop production system in countries with four distinct seasons. Developed countries such as South Korea are becoming more dependent on protective agriculture to support the needs of the growing population. To increase the crop production rate, an optimum environmental condition must be maintained inside the greenhouse through the installation of high-efficiency heating and cooling

systems. A heating and cooling system is used to control the stored heat inside the greenhouse buildings, which is very crucial in maintaining the desired air temperature during both extreme seasons. Approximately 85% of greenhouse owners in Korea use fossil fuel as an energy source for heating to maintain the optimum environment inside the greenhouse facilities [1,2]. Thus, the South Korean government has been strongly promoting alternative ways to reduce the dependence on fossil fuel through the establishment of acts such as the Energy Act and Energy Use Rationalization promoting the use of renewable energy sources for greenhouse crop production. Evidence of this can be seen in the increase of generated renewable energy from 21,751 thousand tonnes of oil equivalent (TOE) in 1995 to 51,427 thousand TOE in 2019, showing an increase of 57.77% [3].

The modern trend in greenhouse energy conservation practice utilized renewable energy to operate the greenhouse. In particular, the wasted heat energy from effluent generated by the thermal or nuclear power plants was being tapped to heat the greenhouse buildings. Thermal effluent refers to the heated seawater used to cool down the engine of the nuclear plant during its operation. During the cooling process, the seawater absorbs a large amount of thermal energy, resulting in a huge amount of energy loss from the power plant. Traditionally, the thermal effluent is 7 °C higher than the average temperature of normal seawater [4]. The current practice of farm owners was to utilize the heat from thermal effluent for heating the greenhouse to maintain the optimum growing environment. As of 2020, a total of 30 units of thermal power plants that were generating a total power of 30,116 MW were strategically placed throughout South Korea, discharging a total of 47.3 billion tons of thermal effluent [5].

As mentioned in many studies, the highest energy consumption and the largest source of environmental impact for greenhouse crop production is accounted for by its heating and cooling systems [6–8]. However, there have been very few studies related to environmental impact assessment of heating and cooling systems in greenhouses since published papers usually relate heating and cooling systems to residential, commercial or industrial buildings and other applications [9–12]. The application of environmental impact tools to assess the burden in a conventional greenhouse was also limited in number. The common research studies concerning greenhouse building structures were usually focused on the environmental impact of crops produced in a controlled environment and were usually compared with the traditional crop production practice such as in open field production. Additionally, the understanding of the qualitative amount of gas emission to the atmosphere of different heating and cooling systems for crop production used in the greenhouse is also inadequate. Thus, a tool capable of qualitatively estimating the amount of gas emitted from the greenhouse structure is deemed important.

The Life Cycle Assessment (LCA) is a potent tool used to calculate the environmental impact caused by the different processes involved in the entire life cycle [13]. During the assessment, the materials and energy flow used during the different product phases (raw material extraction, construction, operation, disposal, etc.) are evaluated in detail. According to Hendricks [14], LCA is capable of identifying the environmental hotspot for different environmental impacts allowing the conservation of energy, carbon and water. The application of LCA has become widespread in the field of food and agriculture in recent years, such as in building construction [15–17], livestock or crop production [18–22]. However, LCA also has main drawbacks, including the dependency on quality and availability of data used affecting the accuracy of the assessment result. Moreover, LCA-related studies only consider the actual energy used during a certain period of production only. Considering the life span of building considered in LCA research studies, information related to energy load for the entire life span may not be available or not properly documented.

This limitation can be solved using the Building Energy Simulation (BES), which is a tool known to estimate the total energy gain and losses through building internal loads such as facility equipment and crops [23–26]. Very often, the BES tool was used to promote energy conservation building design and upgrade building energy code. In spite of the increasing numbers of related studies regarding the use of LCA and building

models, at present, current literature for the integrated BES–LCA is poor due to limited research. The integration of the BES and LCA approaches permits improved assessment of different alternatives that can be used in the system. However, the method, gap and principles of combining these tools at different phases are still not well established. Thus, the final goal of this paper is to discuss the standard practice using LCA and identify the application limitation. Further, it aims to combine BES tools and LCA to facilitate integrated environmental assessment. Lastly, the integrated BES–LCA design approach was used to assess the environmental impact caused by different heating systems in a conventional multi-span Korean greenhouse facility.

## 2. Review of Literatures

*Past BES and LCA Studies*

The application of BES and LCA has been constantly growing in the field of building construction and agriculture. As evidence, several papers have been published related to agriculture. In particular, the buildings, either for commercial, residential or agricultural purposes, have directly or indirectly used energy throughout their life cycle (from manufacturing and operation to decommissioning). Consequently, the building designers and engineers must identify the thermal behaviour of the building to suitably establish retrofitting measures to be adopted in the building construction. Thus, a BES tool that is capable of predicting indoor air temperature needs for heating and cooling is considered necessary. This demand can be resolved using BES software. BES software is a widely used tool that estimates the total energy use of commercial, industrial and agricultural buildings for real-time building control and operation. These tools have also been used by researchers to predict and analyzed the energy demand of building design alternatives for code compliance and green building certification [27–29]. Presently the BES tools used differ in many ways such as the thermodynamic models used, their purpose and their ability to interchange data with other software [30]. In the field of agriculture, the EnergyPlus software was used by Stadler et al. [31] to simulate the onsite generation of an agricultural building. Energy Plus was also used to analyze the energy load of other agricultural and commercial buildings [32,33]. Al-ajmi and Hanby [34] calculated the inside air temperature considering the wall type and window location. Hong et al. [35], on the other hand, used the Transient Systems Simulation Program (TRNSYS) software to analyze the heating load of a naturally ventilated broiler house. Vadiee and Martin [36] analyzed the thermal flow in a large-scale commercial greenhouse using TRNSYS software. Rasheed et al. [25] evaluated the influence of greenhouse building design, including the roof shape, orientation, coverings, etc., on the building energy conservation capacity.

In terms of building environmental impact, literature has usually been subdivided according to the type of application such as heating, ventilation and air conditioning (HVAC) systems [37–41] and greenhouse crop production [42–45]. Further, most of the above-mentioned related studies discussed the three main greenhouse gases contributing to environmental burdens such as the global warming potential ($CO_2$-eq), acidification potential ($SO_2$-eq) and eutrophication potential ($PO_4$-eq) that were known to cause great environmental danger in the long run.

For instance, in the case of heating and cooling systems, Koroneos and Nanaki [46] studied the environmental gas emission of ground source heat pump (GSHP) used for heating and cooling of townhall in Thessalonokki, Greece. The authors concluded that the extraction of the raw materials for HVAC systems was the primary contributor of $CO_2$, $SO_2$ and NOx. This is in contrast with the claim of Saner et al. [47], who state that the operation phase of the system is the highest contributor to gas emission. Carvalho [48] assessed the heating system of a residential house and concluded that when a GSHP was used, a 60% $CO_2$ reduction was predicted by 2050. Blumsack et al. [38] mentioned that GSHP used in a residential house in Pennsylvania decreased $CO_2$ by 62%, increased $SO_2$ by 3% and increased $NO_x$ by 1%, whereas an electrically driven heat pump used in a conventional residential building in Germany was able to reduce the amount of $CO_2$ emission by 45%

compared with an oil broiler [49]. Bayer et al. [50] claimed that the calculated $CO_2$ emission from burning fossil fuels can be reduced by 30% and that all the heating systems in the target region will be replaced. According to Shah et al. [37], the environmental impact of utilizing heat pump systems in a 181 m$^2$ residential house has a lower environmental impact when equated with using a furnace or boiler. The same conclusion was derived in the study of Saner et al. [47] for a 200 m$^2$ residential house. It was further emphasized that about 23% of gas emissions affected the quality of ecosystems. As stated by Bayer et al. [50], the use of coal to operate the GSHP in a residential house contributed to about 80% of the increase in GHG emissions.

Canaj et al. [51] emphasized that in typical crop production in the greenhouse, one of the major environmental factors that affect and contribute to the emission of environmental greenhouse gases was the greenhouse infrastructure. Thus, a number of published papers have focused on the analysis of building structures alone. Similarly, Santos and Costa [41] also claimed that the building was found to contribute to about 40% of the energy consumption and 36% of carbon dioxide. Further, Salehpour et al. [52] studied the different environmental impacts of growing primrose in a greenhouse, showing that about 0.206 kg $CO_2$-eq was emitted per piece of flower produced. Zarei et al. [53] compared the environmental impact of producing cucumber and tomato in a greenhouse and open field. The result of the analysis showed that the use of diesel fuel and natural gas was the major contributor to all impact categories in greenhouse and electricity, and nitrogen-based fertilizers had the highest contribution to all impact categories.

## 3. Materials and Methods

### 3.1. Research Flow

Figure 1 showed the research flow followed for integrating the BES tool with the LCA software. The first steps include the selection of the target experimental greenhouse and case scenarios. The next step is to determine the annual total energy load from the target experimental greenhouse building using BES software. Prior to simulation, the different energy exchange models such as greenhouse, heating and cooling, and crops to predict the annual energy load were modeled and combined. The detailed description of each model was discussed in Lee et al. [54]. The computed annual energy was converted to annual primary energy use by applying annual average conversion factors. The subsequent step includes the LCA analysis to calculate the environmental impact.

### 3.2. Target Experimental Greenhouse

The target experimental greenhouse is located in Chungcheongnam-do Province, South Korea. The facility is located in the western part of Boryeong Power Plant. The Boryeong power plant discharges thermal effluent at around 3 billion tons per year [1]. The greenhouse grows a fixed number of Irwin mangoes and is intended for research purposes only. The greenhouse is divided into two partitions: plant growth room (762 m$^2$) and workroom (128 m$^2$), as shown in Figure 2a. The plant growth room is occupied by 100 potted (7.68 m$^2$ spacing) Irwin mango trees pruned at the height of 1.5 m. To equalize the light interception, these mango trees were structured into a globular shape. The optimum growing environment for the mangoes inside the greenhouse was set at 20 °C. However, to bear fruit, the temperature must be lower than the optimum temperature. The experimental greenhouse has a total of 8 spans with a dimension of 34.2 m × 30.0 m (length × width) with a maximum ridge height of 5.7 m and eave height of 4.5 m (Figure 2b). The greenhouse was covered with 0.15 mm-thick single layer polyolefin film.

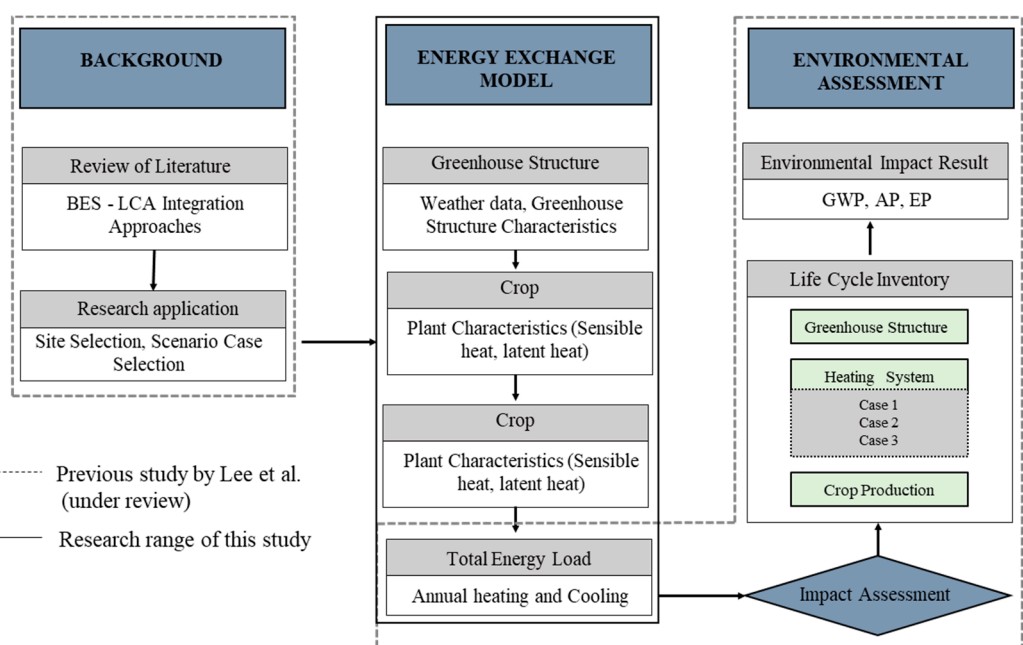

**Figure 1.** Research flow for integrated BES–LCA approach for environmental assessment of greenhouse heating system.

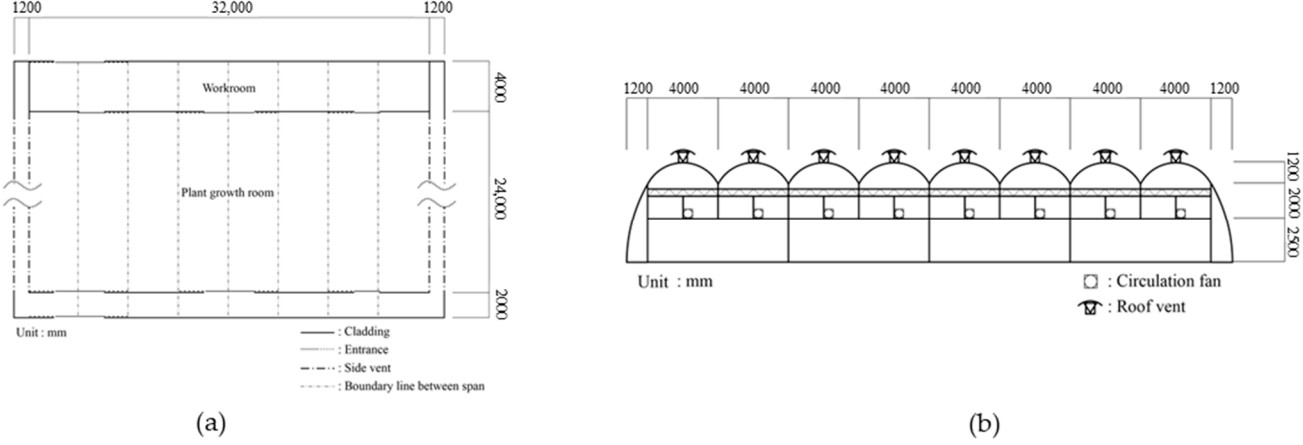

**Figure 2.** (**a**) Description of the greenhouse floor plan; (**b**) front view of structural characteristics of the target greenhouse, (Reprinted with permission from ref. [1]).

Shown in Figure 3 is the different equipment installed inside the greenhouse to facilitate an appropriate growing environmental condition. The structure was equipped with an absorption heat pump system where thermal effluent coming from the Boryeong power plant was used as an energy source for heating the greenhouse. In particular, air ducts and 16 circulation fans with 35 m$^3$/min per unit capacity were strategically installed throughout the building to allow a uniform distribution of heat during the cold season. As can be seen from the figure, the main air duct (60 cm diameter) is directly attached to the heat pump, and the sub-air duct (40 cm diameter) is located near each tree pots.

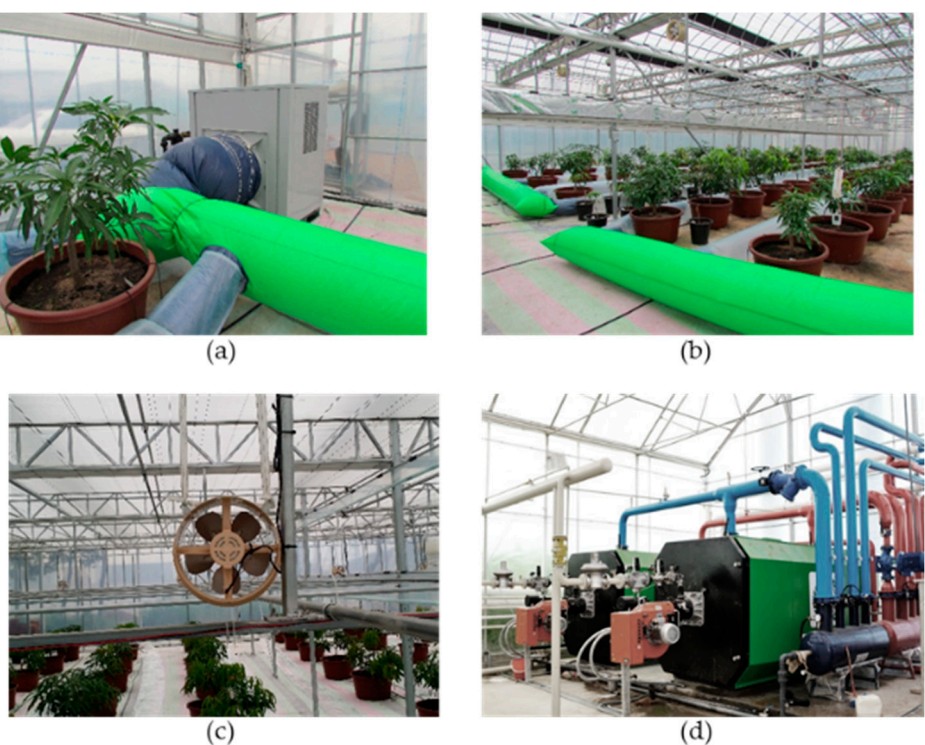

**Figure 3.** Equipment for heat distribution inside the greenhouse (**a**) main air duct, (**b**) sub air duct near crop surface, (**c**) circulation fan and (**d**) heat pump unit.

### 3.3. Softwares and Tools

### 3.3.1. Building Total Energy Demand

In recent years, there was an increase in published researches for utilizing the different BES software for greenhouse buildings [23,26]. The TRNSYS software (Version 18, Solar Energy Laboratory, University of Wisconsin-Madison, Madison, WI, USA) was used to calculate the energy exchange of the experimental greenhouse. TRNSYS refers to commercially available BES software used to predict the energy load of a building. It is a transient simulation software tool where small components such as heat systems can be designed individually and then be combined with the multi-zone building complex. It also offers a wide range of source code and a large component library. Considering this, many energy simulation studies utilized this software for convenience and accuracy.

Since the experimental greenhouse has thin cladding and plants, the target greenhouse was prone to changes in environmental conditions. Therefore, the energy loads were calculated using a dynamic analysis method. As emphasized by Lee et al. [54], a dynamic model refers to a method of calculating the energy load of the building considering the variable change due to the time factor. To calculate the thermal behavior in the experimental greenhouse, the domain was divided into several zones according to Equation (1):

$$Q_i = Qs_{urf} + Q_{inf} + Q_{vent} + Q_{ishcci} + Q_{solar} + Q_{(g,c)} + Q_{cplg} \tag{1}$$

where $Q_i$ is the total heat gain of zone I (kJ), $Q_{surf}$ is the convective heat gain or loss from surfaces (kJ hr$^{-1}$), $Q_{inf}$ is the heat gain or loss by infiltration (kJ hr$^{-1}$), $Q_{vent}$ is the heat gain or loss by ventilation (kJ hr$^{-1}$), $Q_{ishcci}$ is the absorbed solar radiation on all internal shading devices of the zone and directly transferred as a convective gain to the internal air (kJ hr$^{-1}$), $Q_{solar}$ is the fraction of solar radiation entering a zone (kJ hr$^{-1}$), $Q_{(g,c)}$ is the internal convective gains (kJ hr$^{-1}$) and $Q_{cplg}$ is the heat gain or loss due to connective air flow from the adjacent zone (kJ hr$^{-1}$).

However, unlike the expected energy requirement where a lower energy load is required during the early stage of crop production and higher energy load is needed at

the later stage of crops, in this study, the computed annual energy in the greenhouse was assumed to remain constant for the entire life span of the systems. Specifically, as previously mentioned, the temperature requirement for different growth stages of mango differs, resulting in different energy requirements. A total of three dynamic energy exchange models were adopted to estimate the annual building energy load of the greenhouse facility, namely the greenhouse structure, the crop energy exchange and heating systems. Shown in Figure 4 is the combined design of the BES model used for calculating the energy load of the greenhouse

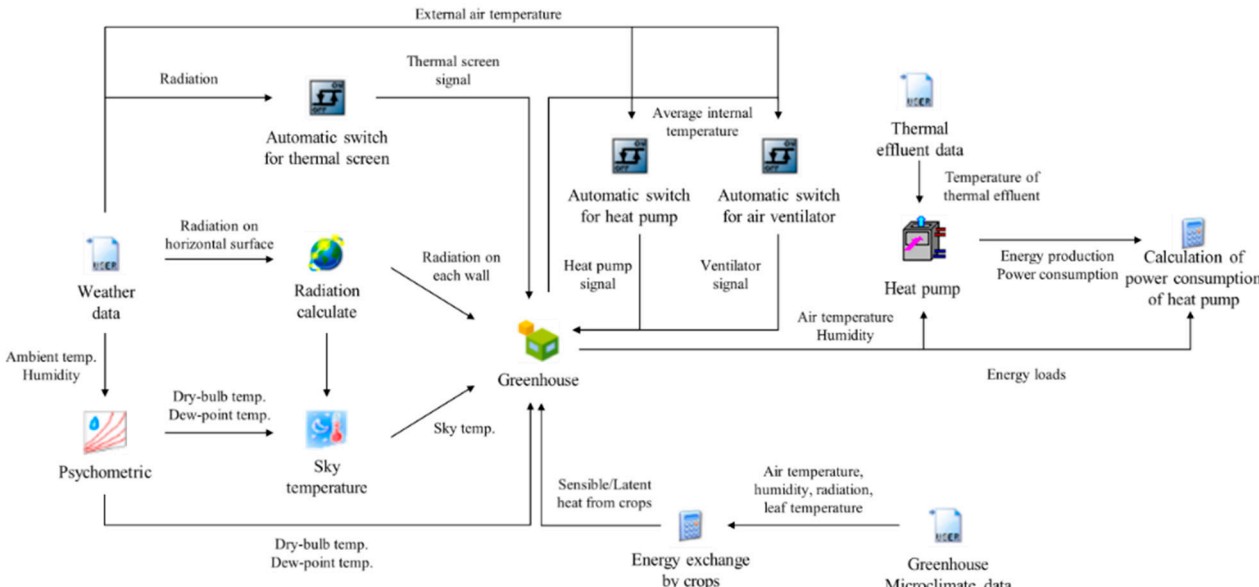

**Figure 4.** Design of BES model for calculating the energy loads of the greenhouse (adapted from [54]).

### 3.3.2. Greenhouse Facility Environmental Impact Assessment

In this study, the current versions of OpenLCA (Version 1.10.3, GreenDelta GmbH, Berlin, Germany) and Ecoinvent database (Version 3.6) were used to analyze the environmental impact of the greenhouse, the heating systems and the crop production materials. LCA follows the ISO standard 14040 up to 14044. As previously defined, LCA is a tool known to quantify the economic burden of inputs and outputs over the entire life cycle. In particular, the LCA study comprised four general phases: the Goal and Scope Definition (ISO 14041), the Inventory Analysis (ISO 14041), the Impact Assessment (ISO 14042) and the Interpretation (ISO 14043). The goal definition determines the purpose of the study, while the scope definition process defines the boundaries of the systems being studied. The inventory analysis was considered to be the most laborious part of the LCA study. In this step, all the major components of the products were listed, and the equivalent unit used was determined either by field experiment data or through a series of literature data. A careful selection of the input inventory data must be done to ensure that all the required input and processes were included in the system. The impact assessment involves the selection of an appropriate inventory assessment method. Under this step, the potential environmental effects of all the processes were considered. Finally, the last step in LCA, where the key findings are presented systematically, was to present the critical sources of impact and the options to reduce these impacts.

The life cycle of a system is typically broken down into five stages: manufacturing, transportation, installation, operation and end-of-life treatment. In terms of economic analysis, the net present value (*NPV*) was adopted. As described by Pombo et al. [55], an energy-efficient building structure could increase energy and cost-saving throughout the entire life cycle. In particular, the author further emphasized that to make a certain product profitable, the energy cost saved over the lifespan must be an investment cost. Thus, in

the calculation, all the costs related to the investment were considered negative and all the energy savings were considered positive. Throughout the economic analysis of this research, the following values were used: the annual inflation rate was set at 4%, annual discount rate was 12%, electricity rate was 40.1 KRW/kWh based on agricultural electricity price, and kerosene price was set at an average of price of 1291 Krw/liter. The *NPV*, which is the difference between the present value of cash inflows and the present value of cash outflows over a period of time, is expressed in Equation (2) as:

$$NPV = \sum_{i=1}^{n} \frac{R_t}{(1+r)^i} - initial\ investment \tag{2}$$

where *NPV* is the net present value, $R_t$ is the net cash inflow during a single period, *i* is equal to the interest rate in present study and t is the number of time periods.

### 3.4. Introduction to Scenario Cases

The initial step in the study is to calculate the total energy load that considers the energy exchange between the greenhouse facility, the heating systems and the crops inside the target experimental greenhouse. Summarized in Table 1 is the computed total energy load using the BES model. A detailed description of the simulation procedure is discussed in Lee et al. [54]. The final stage of this study was to use the TRNSYS BES simulation result values through integration to OpenLCA software and apply the data to the selected scenario cases.

**Table 1.** Annual energy loads of the experimental greenhouse [54].

| Year | Annual | Total Load |
|---|---|---|
| | **Heating (MJ)** | **(kWh)** |
| 2009~2010 | 838,243.37 | 315,960.47 |
| 2010~2011 | 825,247.94 | 312,197.47 |
| 2011~2012 | 802,653.62 | 297,472.02 |
| 2012~2013 | 901,318.95 | 332,306.58 |
| 2013~2014 | 786,186.39 | 306,118.99 |
| 2014~2015 | 813,409.80 | 297,472.27 |
| 2015~2016 | 782,307.87 | 290,323.42 |
| 2016~2017 | 793,454.47 | 300,480.89 |
| 2017~2018 | 813,209.88 | 300,179.06 |
| 2018~2019 | 755,980.29 | 322,996.59 |
| Average | 811,201.26 | 307,550.78 |

- Case 1: Thermal effluent heat-powered absorption heat pump (AHP). When using an AHP as a heating system inside the experimental greenhouse, additional equipment must be installed. This includes but is not limited to the availability of water storage tanks, heat storage tanks and fan coil units. The thermal effluent from the power plant flows into the heat pump inside the greenhouse. From the manufacturers' data, the heat pump has a 43,276 W in maximum cooling capacity and 36,786 W in maximum heating capacity. The energy efficiency (COP) for cooling and heating were 4.68 and 4.61, respectively. Storage tanks of 40 m$^3$ (cold) and 80 m$^3$ (hot) were also constructed to store the water flowing into the system. The fan coil has a power unit of 18,000 (kcal h$^{-1}$) for cooling and 30,000 (kcal h$^{-1}$) for heating. In the environmental assessment, the life span of the heat pump was assumed to be 20 years and was in continuous operation for 24 h a day during the entire winter season. As described in Figure 4, circulation fans and air ducts were installed to maintain the uniformity of heating distribution throughout the whole building.
- Case 2: Electric-powered heat pump. The electric-powered heat pump included in the hypothetical case study with a maximum cooling capacity of 61.6 KW (−5 to 48 °C) and a maximum heating capacity of 69.3 KW (−20 to 24 °C). The unit has COP and

power consumption of 3.55 and 17.35 KW for cooling and 4.15 and 16.70 KW for heating. Similar to the AHP (case 1), the electric-powered heat pump (case 2) used air ducts and fans to uniformly distribute heat throughout the building. Unlike case 1, the case 2 scenario does not require additional installation of facility such as storage tanks unlike those of case 1 since the heat source is electricity. The principle of heating involved in this kind of heat pump allows the heat to move from lower temperature to high temperature. Under this case, the heat transferred in the fan coil provides a higher temperature to the surrounding. The electric-powered heat pump was set to have an expected life span of 15 years.

- Case 3: Kerosene-powered boiler. The second hypothetical case includes the utilization of a natural kerosene-powered boiler. Like the heat pump, the boiler heating system also comprised a various component which includes the burner, chamber, heat exchanger, etc. The basic working principle of a boiler is to store water in a closed vessel and heated by burning fuel in a furnace to produce hot gases. The boiler used in the analysis was assumed to be manufactured abroad with a rated heating of 50 KW and fuel consumption of 5 L/hr. The boiler was also assumed to have 63% operational efficiency. The boiler has a stainless heat exchanger and brass gas burner. Given this material component, the boiler was assumed to be operational for the span of 30 years. Moreover, the kerosene used to power the power was set to have a heating value of 46.20 MJ/kg of kerosene fuel.

### 3.5. Environmental Assessment Process

The following subsections present different phases of an LCA on greenhouse facilities. In this section, the sequential flow of LCA analysis was presented following the four general phases of LCA.

#### 3.5.1. Goal and Scope

As previously mentioned, the goal of the research is to determine the optimum heating systems inside the greenhouse. To reach this goal, two hypothetical case studies (electric-powered system and kerosene-powered boiler) were added in the scenario case and were assumed to be used as a source of heating system of the target greenhouse. The functional unit, which is the basis for the comparative analysis, was defined as 1 $m^2$ of heated and cooled greenhouse floor area for the duration of 1 year. The scope of this work was restricted by omitting all processes that are not related to the function of the greenhouse and in particular those that can be separated from the operation of the greenhouse facilities such as the installation of greenhouse equipment including the microclimate sensors, vents, lightings and fixtures, etc. Moreover, the analysis was performed using a process-based approach wherein the life cycle of the systems is divided into distinct phases: extraction of raw materials, production and disposal.

#### 3.5.2. Data Input Inventory

As previously mentioned, the Ecoinvent 3.6 database was utilized to assess the environmental impact of greenhouse production, heating system and crop production. Accordingly, the dataset used for the greenhouse was set to a 25 years life span and represents the production up to the disposal of a 1 $m^2$ greenhouse with film covering. The building also included a fertigation system, $CO_2$ injection system and storage facilities. For the different heating systems, the datasets for diffusion absorption heat pumps, brine-water heat pump and oil boilers were used. In the case of the energy used, the reduced energy consumption of 59% was assumed considering the study result conducted by Cecconet et al. [54] for heat energy recovery of wastewater. Due to the absence of Irwing mango variety in the Ecoinvent 3.6 database for mango production, this study used a dataset that considered the Tommy Atkins, Palmer Keitt, Kent and Palmar mango varieties instead. These databases were adjusted accordingly to suit the condition of the actual experimental greenhouse. The detailed data inventory was supplemented in Appendix A.

### 3.5.3. Life Cycle Impact Assessment (LCIA)

Technically, most greenhouse gases naturally occur within the Earth's surface; however, the emission was intensified by various human activities, which in turn caused climate change. Therefore, in this study, the CML 2001 method was used to evaluate and compare the impacts of the three heating systems. Specifically, acidification potential (kg $SO_2$-eq), the global warming potential (kg $CO_2$-eq) and eutrophication potential (kg $PO_4$-eq) were used to compare the environmental impact of each case. The AP refers to the different acidifying contaminants, including sulfur dioxide ($SO_2$), nitrogen oxides ($NO_x$) and nitrogen monoxide (NO), that caused acid deposition on both soil and water [56]. On the other hand, Bird et al. [13] and Kumar et al. [57] cited that the $CO_2$-eq, which causes climate change, not only represents $CO_2$ emissions but also represents the non-$CO_2$ greenhouse gases such as methane ($CH_4$) and nitrous oxide ($N_2O$) and have an equivalent factor that is dependent on average residence time in the atmosphere. The same definition and procedure as the GWP was employed when estimating the total PO4-eq emission from the entire life cycle. According to Jan et al. [58], the EP assessed the environmental burden caused by greenhouse gases such as nitrogen (N) and phosphorus (P) to the aquatic and terrestrial ecosystems.

## 4. Results and Discussion

### 4.1. Case Scenario Environmental Analysis and Life Cycle Interpretation

The environmental impact caused by every process in the boundary system was studied by using the inventory dataset. Summarized in Figure 5a is the overall relative indicator results of the simulated scenario cases. As can be seen, the environmental impact caused by the kerosene-powered boiler had the largest contribution among all LCIA criteria. These may be caused by all the output gas emitted during the burning process of kerosene fuel to power the boiler system. The next highest environmental impact was attributed to case 2, which used electricity to provide heat to the entire facility, while case 1 showed the least environmental burden due to less dependency on energy used. Since the main goal of this paper is to analyze the emissions of the major greenhouse gas dispersed into the atmosphere, as previously mentioned, the AP, GWP and EP were the only environmental burdens that are discussed in the following subsections.

Figure 5b, on the other hand, shows the percent contribution of the different systems components used in the assessment. Specifically, it was found that the energy used for heating (source of heat) contributed to the highest environmental burdens with about 43.95% to 96.47, 86.59% to 95.73% and 40.59% to 89.47% for AP, GWP and EP, respectively. However, the construction and maintenance of greenhouse buildings were shown to contribute a maximum of 40.86% when a heat pump was used (case 2) and only 4.94% when a kerosene-powered boiler was utilized. The low contribution of greenhouse building to environmental burdens in case 3 was due to the very high impact of burning kerosene fuel during the operation of the heating systems. Subsequently, the materials used for the production of different heating systems have the least environmental burden at 0.88% to 6.01% contribution, 0.89% to 7.69% contribution, 0.36% to 1.94% contribution for case 1, case 2 and case 3, respectively. This is because the heat pumps and boilers used in the analysis have a long life span. This means that a unit of heat pump or boiler can be used for several years.

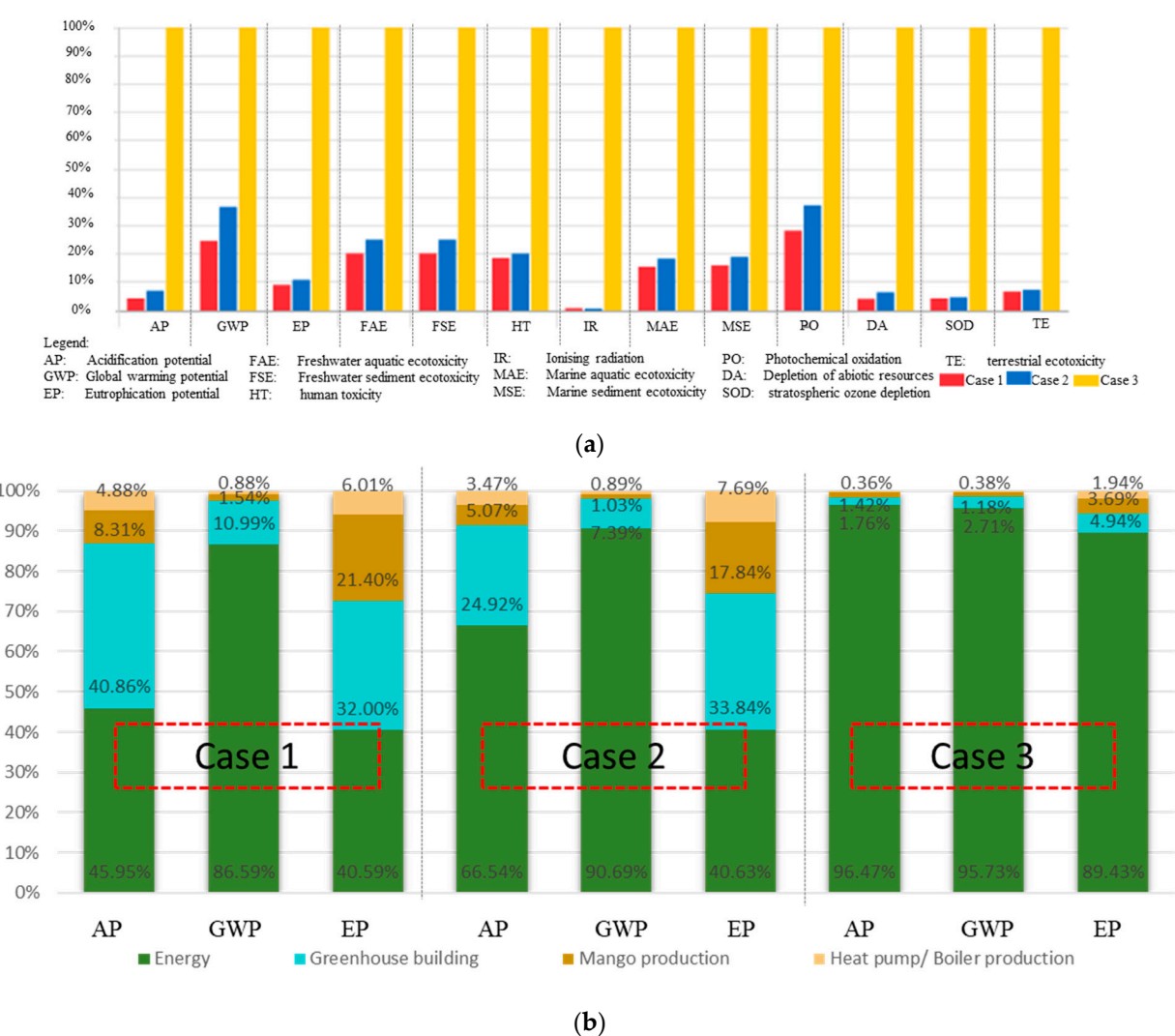

**Figure 5.** (**a**) Detailed LCIA result and (**b**) distribution of environmental category based on system components.

Table 2 shows the total quantitative impact categories of different heating systems used inside the greenhouse. The use of case 1 reduces the AP to $4.954 \times 10^{-2}$ (kg SO$_2$-eq), a very small emission compared to case 3. However, in the case of the GWP in case 1 and case 2, it was greatly reduced from $1.13954 \times 10^2$ kg CO$_2$-eq when case 3 was used. In addition to a higher CO$_2$ emission generated when operating the systems, based on the analysis of the results, the materials used to produce and operate one unit of electricity-powered heat pump also contributed to the high GWP, resulting in a difference in GWP with that of case 1. Lastly, the EP of case 3 was largest by an average of $1.47954 \times 10^{-1}$ kg PO$_4$-eq when compared with case 1 and case 2.

**Table 2.** Total environmental impact of three scenario cases.

|  | AP (kg SO$_2$-eq) | GWP (kg CO$_2$-eq) | EP (kg PO$_4$-eq) |
|---|---|---|---|
| Case 1 | $4.96 \times 10^{-2}$ | $2.79 \times 10^1$ | $1.47 \times 10^{-2}$ |
| Case 2 | $8.12 \times 10^{-2}$ | $4.15 \times 10^1$ | $1.77 \times 10^{-2}$ |
| Case 3 | $1.16 \times 10^0$ | $1.14 \times 10^2$ | $1.63 \times 10^{-1}$ |

Summarized in Table 3 is the detailed quantitative LCIA result of the three case scenarios. From the environmental point of view, it can be observed that the energy used to operate the greenhouse had the highest influence on the assessment result. This is caused

by the different processes involved to generate 1 MJ of energy. In the case of greenhouse building and mango production, constant input and output values were given to each case considering that the heating systems were assumed to be the only factor that changes in the boundary system.

**Table 3.** Detailed impact environmental assessment results in the three scenario cases.

| Impact | Description | Case 1 | Case 2 | Case 3 |
|---|---|---|---|---|
| Acidification Potential (kg $SO_2$-eq) | Energy | $2.28 \times 10^{-2}$ | $5.40 \times 10^{-2}$ | $1.11 \times 10^{-0}$ |
| | Greenhouse building | $2.03 \times 10^{-2}$ | $2.02 \times 10^{-2}$ | $2.02 \times 10^{-2}$ |
| | Mango production | $4.12 \times 10^{-3}$ | $4.11 \times 10^{-3}$ | $1.63 \times 10^{-2}$ |
| | Heat pump/Boiler production | $2.43 \times 10^{-3}$ | $2.81 \times 10^{-3}$ | $4.11 \times 10^{-3}$ |
| Global Warming Potential (kg $CO_2$-eq) | Energy | $2.42 \times 10^{1}$ | $3.75 \times 10^{1}$ | $1.08 \times 10^{2}$ |
| | Greenhouse building | $3.07 \times 10^{0}$ | $3.06 \times 10^{0}$ | $3.06 \times 10^{0}$ |
| | Mango production | $4.29 \times 10^{-1}$ | $4.28 \times 10^{-1}$ | $1.33 \times 10^{0}$ |
| | Heat pump/Boiler production | $2.46 \times 10^{-1}$ | $3.68 \times 10^{-1}$ | $4.28 \times 10^{-1}$ |
| Eutrophication Potential (kg $PO_4$-eq) | Energy | $5.97 \times 10^{-3}$ | $7.16 \times 10^{-3}$ | $1.45 \times 10^{-1}$ |
| | Greenhouse building | $4.71 \times 10^{-3}$ | $5.96 \times 10^{-3}$ | $7.99 \times 10^{-3}$ |
| | Mango production | $3.15 \times 10^{-3}$ | $3.14 \times 10^{-3}$ | $5.96 \times 10^{-3}$ |
| | Heat pump/Boiler production | $8.81 \times 10^{-4}$ | $1.35 \times 10^{-3}$ | $3.14 \times 10^{-3}$ |

The additional potential of LCA is its capability to execute environmental impact contribution analysis on the specific material or process. Presented in Figure 6 are different illustrations showcasing the specific impact contribution of the input materials for the construction and maintenance of the greenhouse building. It must be noted that the materials shown in each figure reflect only the dominant materials causing environmental impact and those with very little contribution were summed and coined as "others". Moreover, those materials with less than 1% contribution were not labeled for better visualization. The top five main contributors of environmental burdens include the use of zinc coating coils, the low alloyed steel used for constructing the greenhouse frame, polyvinyl fluoride and ethyl vinyl which were both used for greenhouse covering and lastly the chromium steel.

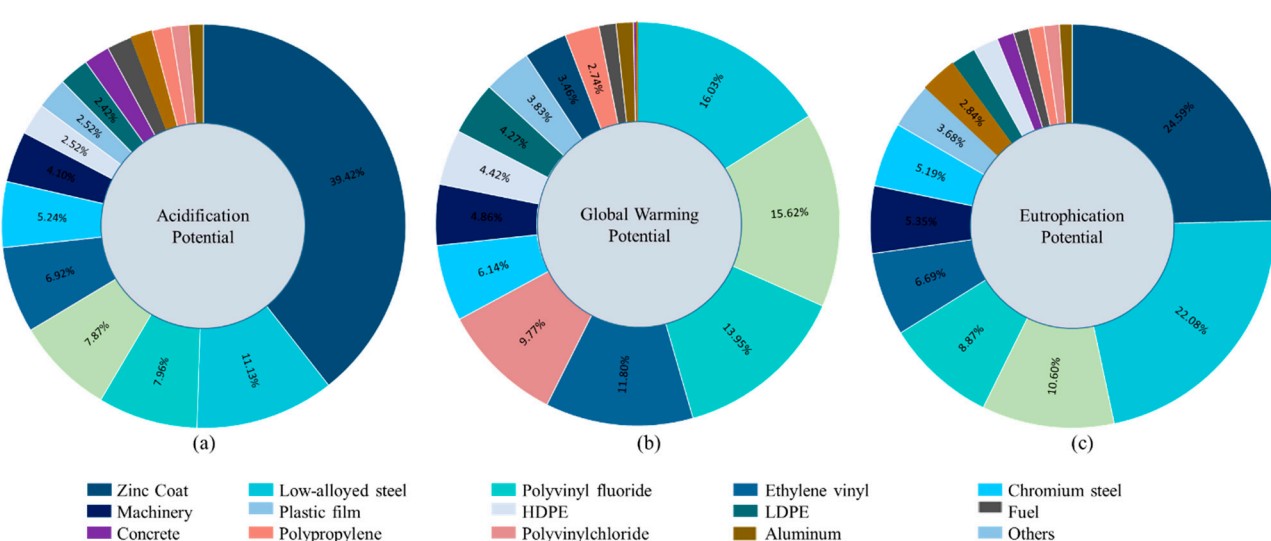

**Figure 6.** Impact distribution for greenhouse building (**a**) AP, (**b**) GWP and (**c**) EP.

### 4.2. Economic Analysis

The target greenhouse was assumed to be used for research purposes, and therefore there was no economic gain during the entire life cycle of both greenhouse building and heating systems. This means that only the cash outflows were included in the NPV

calculation. Ideally, if the NPV has a positive value, it means that the income generated was higher compared with all the accumulated cost. However, a negative NPV indicates that there is an economic loss in the investment. Considering the assumptions made in Section 3.3.2 and the initial cost of KRW 4.4 million for heat pumps and KRW 6.3 million for boiler, the NPV was calculated. The NPV calculation result of different heating systems showed that case 1 has $-120,685,762.44$, case 2 has an NPV of $-131,489,523.75$ and case 3 has $-317,437,668.47$. From this, it can be concluded that utilizing the heat pump system (case 1 and case 2) has a lower negative NPV compared with the use of a boiler. This calculation result accounted for the high maintenance cost of the boiler system, and a continuous hike in the price of kerosene to power the boiler resulted in the highest cash outflows.

*4.3. Discussion on the Comparison of Different Heating Systems*

The choice between a heat pump (case 1 or case 2) and boilers (case 3) should be made based on what is most important to the farm owner, the budget and the location of the greenhouse farm. When comparing the physical properties of different heating systems, case 1 and case 2 are effective and efficient heating systems due to high heating efficiency. As cited in many studies, many older boilers are only low heating efficiency of 50–75%. This means that a large amount of energy used for the operation was not utilized and wasted. As claimed by several authors and manufacturing companies, the current boilers available on the market can achieve a high efficiency of 92% where only 8% of energy is wasted. Accordingly, to attain a high-efficiency boiler, it was suggested to replace the oil boilers used with a new modern model. However, the drawback of utilizing case 1 and case 2 as heating as a heating option is its lower lifespan which only lasted for about 15 to 20 years compared with boilers, which can last up to 30 years. Secondly, the installation of case 1 and case 2 required the utilization of outside space. This is specifically true for case 1, which needs to have a larger space for the construction of a hot and cold water storage tank, which adds to the initial economic cost. For case 2, a smaller outside space was needed, and in some cases, there will also be an indoor unit including the heat exchanger. In case 3, a typical kerosene-powered boiler is considerably more compact compared with the previously mentioned heating systems.

One of the goals of this paper is to assess the environmental impact caused by different heating systems used in a greenhouse. The analysis revealed that the use of an electric-powered heat pump (case 2) resulted in a higher environmental impact when compared with the use in case 1 of thermal effluent heat-powered AHP). However, this result may be different if the source of electricity is obtained from renewable sources such as wind, solar, tidal or hydropower energy source. The main factor that was found to give this assessment result is that a higher amount of heat energy from electric sources was being utilized to operate the system. Moreover, it was also found that case 3 had the highest environmental burden. This is agreed by the conclusion stated by [37,49,58] which stated that a boiler has the highest carbon footprint compared to a different type of heat pumps systems. Especially, according to Greening and Azapagic [59], in the case of GWP, heat pumps can save up to 36% of $CO_2$-eq. Nevertheless, the case 1 and case 2 systems have higher efficiency, which makes them the far more environmentally friendly choice.

In the case of the economic aspect, although heat pumps use electricity to run, which is around four times the price of gas, the fact that the heat pump is so efficient means that it uses very little electricity and the running costs are therefore comparable. An emphasis should also be given to the use of thermal effluent heat-powered AHP (case 1) as a source of heat since the amount of heat extracted from the thermal effluent greatly reduces the amount of energy required to maintain the optimum environment of the target greenhouse. For case 3, it was shown that a lower investment cost was possible using this heating system. However, due to over-exploitation of fossil fuels like gas and oil continuously resulting in energy resource depletion, it is likely that these prices will continue to rise in the future, resulting in a higher economic burden to farm owners.

## 5. Conclusions

This study aims to assess and compare both the environmental and economic impact of different heating systems typically used in conventional Korean greenhouse facilities. In the first part of the research, the total energy load needed for crop production on the target experimental greenhouse was calculated using the Building Energy Simulation (BES) software. The average total annual heating load obtained from the calculation was used as the reference for the heating requirement for the life cycle analysis. Three scenario cases were analyzed in the study, which included case 1 (thermal effluent heat powered absorption heat pump (AHP)), case 2 (electric powered heat pump) and case 3 (kerosene powered boiler). The OpenLCA free source software and Ecoinvent 3.6 database was used in the study. The result showed that the use of case 3 as a heating source offers significant environmental disadvantages. Specifically, the environmental assessment revealed that the environmental impact caused by this system is largest in terms of the acidification potential (AP), global warming potential (GWP) and Eutrophication Potential (EP) of $1.15 \times 10^0$ kg $SO_2$-eq, $1.13 \times 10^2$ kg $CO_2$-eq and $1.62 \times 10^{-1}$ kg $PO_4$-eq, respectively. Among the three cases, the thermal effluent heat-powered AHP was found to have a lower environmental burden. Specifically, the AP of case 1 was 38.99 to 95.70%, GWP was 32 to 76% to 75.33% and EP was 16.63% to 90.92% lower compared with case 2 and case 3. Detailed analysis of the results showed that the main contributor to greenhouse gas emission was caused by the type, amount and source of energy used to heat the greenhouse, which contributed to a maximum of 86.59% for case 1, 96.69% for case 2 and a maximum of 96.47% for case 3, depending on the type of gas being considered. The contribution of greenhouse gas emissions caused by building construction, operation and maintenance can also contribute to up to 40.86% of the environmental burden. Finally, the economic analysis of three cases showed that case 1 tends to give a lesser economic burden compared with the other two cases. The finding obtained from this study can be used to support decision making on the selection of the appropriate heating system to be used in the greenhouse. However, further evaluation is mandated considering other types of heating systems typically used in the greenhouse.

**Author Contributions:** Conceptualization, C.D.-V., U.-H.Y., S.-Y.L.; Methodology, C.D.-V.; software, C.D.-V.; data curation, C.D.-V., S.-Y.L.; writing—original draft preparation, C.D.-V., writing—review and editing, I.-B.L., U.-H.Y.; S.-Y.L.; J.-G.K., S.-J.P., Y.-B.C., J.-H.C., visualization, H.-H.J.; supervision, I.-B.L.; All authors have read and agreed to the published version of the manuscript.

**Funding:** This research received no external funding.

**Institutional Review Board Statement:** Not applicable.

**Informed Consent Statement:** Not applicable.

**Data Availability Statement:** All the data used in the manuscript was attached in the appendix table. This dataset was extracted on ecoinvent database (mentioned in the materials and method chapter) and modified according to our selected functional unit.

**Conflicts of Interest:** The authors declare no conflict of interest.

## Appendix A

**Table A1.** Material inventory data for the Greenhouse.

| Flow | Amount | Unit |
|---|---|---|
| **Input** | | |
| acrylic varnish, without water, in 87.5% solution state | $1.39 \times 10^{-3}$ | kg |
| agricultural machinery, unspecified | $9.50 \times 10^{-3}$ | kg |
| aluminum scrap, post-consumer | $-8.00 \times 10^{-3}$ | kg |
| aluminum, cast alloy | $8.00 \times 10^{-3}$ | kg |
| bitumen seal | $1.25 \times 10^{-4}$ | kg |

**Table A1.** *Cont.*

| Flow | Amount | Unit |
|---|---|---|
| blow molding | $3.96 \times 10^{-3}$ | kg |
| calendering, rigid sheets | $1.25 \times 10^{-3}$ | kg |
| concrete block | $4.17 \times 10^{-2}$ | kg |
| concrete block | $9.20 \times 10^{-2}$ | kg |
| copper | $8.10 \times 10^{-4}$ | kg |
| diesel, burned in building machine | $4.67 \times 10^{-1}$ | MJ |
| electricity, low voltage | $2.35 \times 10^{-3}$ | kWh |
| electronics, for control units | $2.00 \times 10^{-5}$ | kg |
| ethylene vinyl acetate copolymer | $1.53 \times 10^{-1}$ | kg |
| extrusion, plastic film | $2.12 \times 10^{-1}$ | kg |
| extrusion, plastic pipes | $9.28 \times 10^{-2}$ | kg |
| glass-fiber-reinforced plastic, polyamide, injection molded | $1.02 \times 10^{-4}$ | kg |
| injection molding | $2.95 \times 10^{-2}$ | kg |
| iron scrap, unsorted | $-3.51 \times 10^{-1}$ | kg |
| polycarbonate | $1.25 \times 10^{-3}$ | kg |
| polyester resin, unsaturated | $1.71 \times 10^{-3}$ | kg |
| polyethylene, high-density, granulate | $5.99 \times 10^{-2}$ | kg |
| polyethylene, linear low density, granulate | $5.95 \times 10^{-2}$ | kg |
| polymer foaming | $6.13 \times 10^{-3}$ | kg |
| polypropylene, granulate | $3.77 \times 10^{-2}$ | kg |
| polystyrene, expandable | $6.13 \times 10^{-3}$ | kg |
| polyvinylfluoride | $2.99 \times 10^{-2}$ | kg |
| section bar extrusion, aluminium | $8.00 \times 10^{-3}$ | kg |
| section bar rolling, steel | $2.82 \times 10^{-1}$ | kg |
| sheet rolling, steel | $4.44 \times 10^{-2}$ | kg |
| silicone product | $1.50 \times 10^{-4}$ | kg |
| steel, chromium steel 18/8 | $4.30 \times 10^{-2}$ | kg |
| steel, low-alloyed | $3.07 \times 10^{-1}$ | kg |
| synthetic rubber | $3.75 \times 10^{-4}$ | kg |
| tractor, 4-wheel, agricultural | $1.90 \times 10^{-2}$ | kg |
| wire drawing, copper | $2.25 \times 10^{-3}$ | kg |
| zinc coat, coils | $2.93 \times 10^{-2}$ | m$^2$ |
| **Output** | | |
| waste concrete | $1.19 \times 10^{-0}$ | kg |
| waste electric and electronic equipment | $8.30 \times 10^{-4}$ | kg |
| waste plastic, mixture | $1.41 \times 10^{-3}$ | kg |
| waste polyvinylchloride | $1.07 \times 10^{-4}$ | kg |
| waste rubber, unspecified | $3.60 \times 10^{-4}$ | kg |

**Table A2.** Material inventory data for mango production.

| Flow | Amount | Unit |
|---|---|---|
| **Input** | | |
| application of plant protection product, by field sprayer | $4.51 \times 10^{-3}$ | ha |
| carbon dioxide, in air | $4.01 \times 10^{0}$ | kg |
| chlorine dioxide | $3.45 \times 10^{-7}$ | kg |
| cobalt | $1.09 \times 10^{-4}$ | kg |
| dolomite | $5.23 \times 10^{-3}$ | kg |
| energy, gross calorific value, in biomass | $3.26 \times 10^{1}$ | MJ |
| ethoxylated alcohol (AE > 20) | $6.33 \times 10^{-5}$ | kg |
| gypsum, mineral | $1.66 \times 10^{-1}$ | kg |
| harvesting, forestry harvester | $1.25 \times 10^{-4}$ | h |
| irrigation | $1.07 \times 10^{-2}$ | m$^3$ |
| lime | $2.62 \times 10^{-4}$ | kg |

**Table A2.** *Cont.*

| Flow | Amount | Unit |
|---|---|---|
| magnesium oxide | $2.96 \times 10^{-2}$ | kg |
| mancozeb | $1.73 \times 10^{-4}$ | kg |
| manganese concentrate | $1.39 \times 10^{-3}$ | kg |
| mango seedling, for planting | $4.71 \times 10^{-3}$ | Item(s) |
| molybdenum trioxide | $6.35 \times 10^{-5}$ | kg |
| nitrogen fertilizer, as N | $3.70 \times 10^{-2}$ | kg |
| occupation, permanent crop, irrigated | $3.76 \times 10^{0}$ | $m^2 * a$ |
| packaging, for fertilizers | $7.91 \times 10^{-1}$ | kg |
| packaging, for pesticides | $5.49 \times 10^{-2}$ | kg |
| pesticide, unspecified | $2.71 \times 10^{-3}$ | kg |
| phenol | $1.98 \times 10^{-5}$ | kg |
| phosphate fertiliser, as $P_2O_5$ | $2.64 \times 10^{-2}$ | kg |
| planting with starter fertilizer, by no-till planter | $1.25 \times 10^{-5}$ | ha |
| polydimethylsiloxane | $1.15 \times 10^{-5}$ | kg |
| potassium fertiliser, as $K_2O$ | $7.71 \times 10^{-2}$ | kg |
| sulfur | $1.93 \times 10^{-2}$ | kg |
| tap water | $4.26 \times 10^{-5}$ | kg |
| tillage, harrowing, by offset leveling disc harrow | $2.50 \times 10^{-5}$ | ha |
| tillage, subsoiling, by subsoiler plow | $1.25 \times 10^{-5}$ | ha |
| transformation, from permanent crop, irrigated | $1.88 \times 10^{-1}$ | $m^2$ |
| weed control, by brush cutter, pasture | $2.25 \times 10^{-3}$ | ha |
| zinc oxide | $3.66 \times 10^{-3}$ | kg |
| **Output** | | |
| Abamectin | $1.58 \times 10^{-6}$ | kg |
| Ammonia | $3.68 \times 10^{-3}$ | kg |
| Azoxystrobin | $7.32 \times 10^{-5}$ | kg |
| Cadmium | $1.13 \times 10^{-6}$ | kg |
| Cadmium, ion | $1.58 \times 10^{-8}$ | kg |
| Cadmium, ion | $3.53 \times 10^{-9}$ | kg |
| Carbon dioxide, fossil | $2.55 \times 10^{-2}$ | kg |
| Chloride | $3.45 \times 10^{-7}$ | kg |
| Chromium | $9.36 \times 10^{-6}$ | kg |
| Chromium, ion | $7.40 \times 10^{-6}$ | kg |
| Chromium, ion | $3.92 \times 10^{-7}$ | kg |
| Copper | $2.40 \times 10^{-12}$ | kg |
| Copper, ion | $1.12 \times 10^{-6}$ | kg |
| Copper, ion | $2.92 \times 10^{-7}$ | kg |
| Difenoconazole | $6.61 \times 10^{-5}$ | kg |
| Dinitrogen monoxide | $8.95 \times 10^{-4}$ | kg |
| Ethephon | $9.65 \times 10^{-5}$ | kg |
| Indoxacarb | $1.19 \times 10^{-5}$ | kg |
| Lead | $4.48 \times 10^{-7}$ | kg |
| Mancozeb | $1.73 \times 10^{-4}$ | kg |
| Nickel | $1.98 \times 10^{-6}$ | kg |
| Nitrate | $6.80 \times 10^{-2}$ | kg |
| Nitrogen oxides | $1.46 \times 10^{-3}$ | kg |
| Pesticides, unspecified | $1.94 \times 10^{-3}$ | kg |
| Phosphorus | $1.67 \times 10^{-5}$ | kg |
| Pyraclostrobin (prop) | $3.67 \times 10^{-5}$ | kg |
| Spinosad | $5.48 \times 10^{-8}$ | kg |
| Tebuconazole | $1.06 \times 10^{-4}$ | kg |
| Thiophanat-methyl | $3.63 \times 10^{-4}$ | kg |
| Trifloxystrobin | $1.05 \times 10^{-5}$ | kg |
| waste wood, untreated | $4.58 \times 10^{-1}$ | kg |
| Water | $3.11 \times 10^{0}$ | $m^3$ |
| Zinc | $1.08 \times 10^{-5}$ | kg |
| Zinc, ion | $5.76 \times 10^{-6}$ | kg |

**Table A3.** Material inventory data for thermal effluent heat source AHP (Case 1).

| Flow | Amount | Unit |
|---|---|---|
| **Input** | | |
| aluminum, wrought alloy | $4.30 \times 10^{-3}$ | kg |
| ammonia, liquid | $2.93 \times 10^{-4}$ | kg |
| building, hall, steel construction | $1.74 \times 10^{-6}$ | m$^2$ |
| building, multi-story | $1.04 \times 10^{-5}$ | m$^3$ |
| copper | $9.77 \times 10^{-4}$ | kg |
| electricity, low voltage | $7.81 \times 10^{-3}$ | kWh |
| electricity, medium voltage | $2.60 \times 10^{-2}$ | kWh |
| electronics, for control units | $1.95 \times 10^{-5}$ | kg |
| helium | $7.81 \times 10^{-4}$ | kg |
| injection moulding | $3.13 \times 10^{-4}$ | kg |
| Occupation, industrial area, built up | $7.81 \times 10^{-4}$ | m$^2$ * a |
| polyethylene, high density, granulate | $3.22 \times 10^{-2}$ | kg |
| reinforcing steel | $6.25 \times 10^{-3}$ | kg |
| sheet rolling, chromium steel | $3.22 \times 10^{-2}$ | kg |
| sheet rolling, steel | $6.25 \times 10^{-3}$ | kg |
| steel, chromium steel 18/8, hot rolled | $1.56 \times 10^{-3}$ | kg |
| stone wool, packed | $6.25 \times 10^{-6}$ | kg |
| Transformation, from unknown | $6.25 \times 10^{-6}$ | m$^2$ |
| Transformation, to industrial area, built up | $3.91 \times 10^{-2}$ | m$^2$ |
| tube insulation, elastomere | $7.81 \times 10^{-4}$ | kg |
| water, completely softened | $4.18 \times 10^{-4}$ | kg |
| water, completely softened | $5.59 \times 10^{-4}$ | kg |
| Water, unspecified natural origin | $5.96 \times 10^{-5}$ | m$^3$ |
| zinc coat, coils | $2.93 \times 10^{-3}$ | m$^2$ |
| **Output** | | |
| electronics scrap from control units | $7.81 \times 10^{-4}$ | kg |
| waste mineral wool | $1.11 \times 10^{-3}$ | kg |
| waste mineral wool | $4.55 \times 10^{-4}$ | kg |
| waste polyethylene/polypropylene product | $4.41 \times 10^{-4}$ | kg |
| waste polyethylene/polypropylene product | $1.12 \times 10^{-3}$ | kg |
| wastewater, from residence | $1.99 \times 10^{-5}$ | m$^3$ |

**Table A4.** Material inventory data for electric heat pump (Case 2).

| Flow | Amount | Unit |
|---|---|---|
| **Input** | | |
| copper | $5.73 \times 10^{-3}$ | kg |
| electricity, medium voltage | $3.65 \times 10^{-2}$ | kWh |
| lubricating oil | $4.43 \times 10^{-4}$ | kg |
| polyvinylchloride, bulk polymerized | $2.60 \times 10^{-4}$ | kg |
| refrigerant R134a | $8.05 \times 10^{-4}$ | kg |
| reinforcing steel | $1.95 \times 10^{-2}$ | kg |
| steel, low-alloyed, hot rolled | $5.21 \times 10^{-3}$ | kg |
| tube insulation, elastomere | $2.60 \times 10^{-3}$ | kg |
| Water, unspecified natural origin | $1.84 \times 10^{-4}$ | m$^3$ |
| **Output** | | |
| ethane, 1,1,1,2-tetrafluoro-, HFC-134a | $9.20 \times 10^{-1}$ | kWh |
| waste plastic, mixture | $1.59 \times 10^{1}$ | MJ |
| water | $1.42 \times 10^{-1}$ | m$^3$ |
| water | $8.02 \times 10^{-1}$ | m$^3$ |

**Table A5.** Material inventory data for kerosene powered boiler (Case 3).

| Flow | Amount | Unit |
|---|---|---|
| **Input** | | |
| alkyd paint, white, without solvent, in 60% solution state | $2.17 \times 10^{-3}$ | kg |
| aluminum, cast alloy | $1.30 \times 10^{-2}$ | kg |
| brass | $4.34 \times 10^{-5}$ | kg |
| brazing solder, cadmium free | $5.21 \times 10^{-3}$ | kg |
| copper | $2.17 \times 10^{-2}$ | kg |
| corrugated board box | $1.01 \times 10^{-1}$ | kg |
| corrugated board box | $8.58 \times 10^{-3}$ | kg |
| polyethylene, high density, granulate | $1.22 \times 10^{-3}$ | kg |
| steel, chromium steel 18/8, hot rolled | $2.17 \times 10^{-2}$ | kg |
| steel, low-alloyed, hot rolled | $4.21 \times 10^{-1}$ | kg |
| stone wool, packed | $1.65 \times 10^{-2}$ | kg |
| tap water | $6.43 \times 10^{-1}$ | kg |
| waste paperboard, unsorted | $-8.68 \times 10^{-3}$ | kg |
| **Output** | | |
| hazardous waste, for incineration | $1.69 \times 10^{-3}$ | kg |
| hazardous waste, for incineration | $3.52 \times 10^{-3}$ | kg |
| waste mineral wool, for final disposal | $9.39 \times 10^{-3}$ | kg |
| waste mineral wool, for final disposal | $7.10 \times 10^{-3}$ | kg |
| waste plastic, mixture | $1.35 \times 10^{-4}$ | kg |
| waste plastic, mixture | $9.46 \times 10^{-4}$ | kg |
| waste plastic, mixture | $1.15 \times 10^{-4}$ | kg |
| waste plastic, mixture | $3.73 \times 10^{-6}$ | kg |
| waste plastic, mixture | $6.83 \times 10^{-7}$ | kg |
| waste plastic, mixture | $9.03 \times 10^{-6}$ | kg |
| waste plastic, mixture | $3.54 \times 10^{-6}$ | kg |
| waste plastic, mixture | $1.51 \times 10^{-6}$ | kg |
| wastewater from pig iron production | $6.22 \times 10^{-5}$ | m$^3$ |
| wastewater from pig iron production | $4.72 \times 10^{-4}$ | m$^3$ |
| Water | $9.65 \times 10^{-5}$ | m$^3$ |
| Water | $1.29 \times 10^{-5}$ | m$^3$ |

**Table A6.** Inventory for energy source used for heating the greenhouse.

| Flow | Amount | Unit |
|---|---|---|
| **Input** | | |
| heat, district or industrial, natural gas | $4.34 \times 10^{2}$ | MJ |
| heat, district or industrial, natural gas | $1.06 \times 10^{3}$ | MJ |
| kerosene | $2.29 \times 10^{1}$ | kg |

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
