# Peer review of "Integrated Building Energy Simulation–Life Cycle Assessment (BES–LCA) Approach for Environmental Assessment of Agricultural Building: A Review and Application to Greenhouse Heating Systems"

_agronomy, doi:10.3390/agronomy11061230_

Round 1

Reviewer 1 Report

Dear authors, 

The present paper is written nicely by taking care of novelties, but some minor corrections are needed to make it perfect, kindly see my suggestions below:

Minor comments:

Line line of abstract 32-33: could be written like ....................Moreover, the thermal effluent heated 31 heat pump was found to emit a low amount of SO2 as SO2-eq (0.46 kg), CO2 as CO2-eq (105.03 kg), and PO4 as PO4-eq (0.15 kg).

Line number 51: kindly define the abbreviation before using it first time (e.g., TOE, TRNSYS, HVAC)

Line no 78-79: need to be support (kindly support with past literature)

Line no 122: need to be cite as per journal format (e.g., Stadler et al. (2006); line no 193; Becalli et al. (2016))

Line no 134-136: meaning not clear, kindly remove or edit (kindly see these two references : https://www.sciencedirect.com/science/article/abs/pii/S0959652619320050)

Line no 163 : 36% of carbon dioxide CO2 (kindly remove carbon dioxide)

Authors has to define CO2-eq. (see this paper and cite; https://link.springer.com/article/10.1007/s00477-018-1608-z)

Figure 3 should be shifted in supplement figure (Figure S1)

Equation number has to be define like Eq 1, 2, 3 ...

Sub-script and super-script need to be check carefully (e.g., line no 264)

References need to updated and arrange as per the journal format 

Author Response

Response: 

Thank you for taking the time to review our manuscript. We appreciated your technical support to improve the content of this research paper. We have carefully read and corrected the insufficient parts of the manuscript that requires revision and improvements. We tried to respond to all the comments made by the reviewers as best as we can. Also, it was stated from the review that some fine grammatical corrections and typos in the paper need to be checked and corrected. For this matter, we have tried our best to identify all the grammatical errors within the manuscript and improve its content for it to be easily understood. We hope that the responses in each inquiry were satisfactory.

Line line of abstract 32-33: could be written like ....................Moreover, the thermal effluent heated 31 heat pump was found to emit a low amount of SO2 as SO2-eq (0.46 kg), CO2 as CO2-eq (105.03 kg), and PO4 as PO4-eq (0.15 kg).

CDV: The abstract was improved as advised by the two authors. Thus the suggestion made by reviewer 1 about line 32 to 33 was fully not incorporated in the revised paper. However, I see to it that the way the abstract in the revised manuscript was clear and well written. See line 30 to 33 for example.

Line number 51: kindly define the abbreviation before using it first time (e.g., TOE, TRNSYS, HVAC)

CDV: Thank you for the comment. All the abbreviation that was mentioned first in the manuscript was defined first and was abbreviated in the next sentences where it was mentioned. For instance: See line 55-56 for TOE, line 133 for TRNSYS,  line 140 to 141 for HVAC,

Line no 78-79: need to be support (kindly support with past literature)

CDV: The comment on this sentence is quite similar to the comment made by reviewer 2. Which  I answered like this:

“Thank you for pointing this out. I miswrote this sentence. What I meant by this sentence is that at present, there were only very few studies about the life cycle assessment for different heating and cooling systems in greenhouses. I revised the content like this:

(Line 70-75) However, there were very few studies related to environmental impact assessment of heating and cooling systems in greenhouses since published paper usually relate heating and cooling systems to residential, commercial and industrial buildings.”

In addition, I inserted some references (reference 10 to 13) I found related to this statement to support my claim (as suggested by reviewer 1).

Line no 122: need to be cite as per journal format (e.g., Stadler et al. (2006); line no 193; Becalli et al. (2016))

CDV: In the revised manuscript, the format and references of all the authors mentioned in the manuscript were carefully checked. All the mentioned references were cited in accordance with the MPDI agronomy journal format.

Line no 134-136: meaning not clear, kindly remove or edit (kindly see these two references: https://www.sciencedirect.com/science/article/abs/pii/S0959652619320050)

CDV: Thank you for the suggestion. In line 141 to 145 previously 134 to 136, I revised like this:

“Further, most of the above mentioned related studies discussed the three main greenhouse gases contributing to environmental burdens such as the global warming potential (CO2), acidification potential (SO2) and eutrophication potential (PO4) that were known to cause great environmental danger in the long run.”

What I intend to emphasize here is that among the literature cited in the previous sentence line (140 to 141), most of the authors only considers 3 major greenhouse gases which include the CO2, SO2 and PO4 gas emission. Referring to the selected main gas emission from literature, in this manuscript, we limit our research result to the three main gases contributing to the environmental burden.

Line no 163 : 36% of carbon dioxide CO2 (kindly remove carbon dioxide)

CDV: I fully support your comment. The CO2 was deleted from the manuscript to avoid redundancy.

Authors has to define CO2-eq. (see this paper and cite; https://link.springer.com/article/10.1007/s00477-018-1608-z)

CDV: From the literature that I have read on life cycle assessment studies, it is very unusual to define in detail the meaning of CO2-eq and other impact categories for Life cycle inventory assessment (LCIA) as it was assumed that these parameters are basic information/knowledge for the assessment especially those that were using LCA software. This is the reason why in our initial manuscript we have not discussed these in categories detail. 

However, to incorporate the reviewer's suggestion, I have included an additional statement in Line  375 to 390. Under this paragraph, I have discussed the different contaminant under each impact categories and have briefly talked about how non-CO2 gases were converted in CO2-eq. Further, instead of only focussing on the definition of CO2-eq, I have included a brief description of SO4-eq and PO4-eq.

Figure 3 should be shifted in supplement figure (Figure S1)

CDV: I appreciated your suggestion. Actually, prior to the initial submission of the manuscript, we thought of transferring the said image (Figure 3) to the supplement section. However, considering that there is only 1 image to be including in that section, we thought that this specific figure will be put in materials and method. In this way, readers can grasp the actual scenario in our selective experimental greenhouse without going to the bottom section of the manuscript where supplement materials and tables were inserted. Thus, in our revised manuscript, we retained Figure 3 under material and methods.

Equation number has to be define like Eq 1, 2, 3 ...

CDV:  Thank you for the comment. All the equations present in the manuscript were defined according to the reviewer’s suggestion

Sub-script and super-script need to be check carefully (e.g., line no 264)

CDV: Sub-script and super-script were checked especially for CO2, SO2, PO4, m2, m3, °C

References need to updated and arrange as per the journal format 

CDV: references were carefully checked and updated.

Reviewer 2 Report

Review. Round one.

The article entitled "Integrated BES-LCA approach for environmental assessment of agricultural building: a review and application to greenhouse heating systems" proposes to study, through the combination of BES and LCA techniques, the equivalent CO2, SO2 and PO4 emissions of three heating systems of greenhouses.

The work and the conclusions are interesting, although the end result is quite predictable.

For your publication, I think it would be interesting to make some improvements to the article:

Major Flaws:

In the discussion section, I miss the comparison of the results obtained by the authors with other previous studies that have been cited in the review of the literatures, such as “Vadiee, A. and V. Martin, Energy analysis and thermoeconomic assessment of the closed greenhouse - The largest commercial solar building. Applied Energy, 2013. 102: p. 1256-1266 "or" Ha, T. and I.K. Lee, Kyeongseok Kwon, Hong, Sewoon, Computation and field experiment validation of greenhouse energy load using building energy simulation model. Int J Agric & Biol Eng, 2015. "

  • What are the differences between the results obtained by these studies and those obtained in this article?
  • What reasons do you think produce these differences?
  • are these differences congruent?
  • Are the results obtained in this article contradictory with those previously published by some other researcher?

Minor Flaws:

  • It is not usual to include acronyms in the title, except in those cases in which they are widely known in the scientific field.
  • Lines 67-68: this paragraph corresponds to references 7-9, while the next paragraph, lines 69-70 would correspond to reference 6.
  • Lines 75-77. The authors state that “the qualitative amount of gas emission to the atmosphere of different heating and cooling system for crop production used in the greenhouse is also inadequate”. Why?
  • Line 475: “This is agreed by the conclusion stated by Toledo et al. (2019) and Samari et al., (2019) which stated that… ”Reference numbers must be included and the results obtained by these authors must also be included in section 2. Review of literatures.
  • Line 165: “carbon dioxide CO2”. You have to eliminate CO2 because it is repetitive.
  • Line 193: correct “mongo” by “mango”
  • Line 217: reference 26 does not correspond to studies carried out in greenhouses, so it should be eliminated.
  • Line 267: correct “appropriate” by “appropiate”
  • Line 321: correct “in the fan coal” by “in the fan coil”
  • Line 360: You must include the reference and its number: "the study result conducted by Cecconet et al in 2020 for heat ..."

Author Response

Reviewer 2

Review. Round one.

The article entitled "Integrated BES-LCA approach for environmental assessment of agricultural building: a review and application to greenhouse heating systems" proposes to study, through the combination of BES and LCA techniques, the equivalent CO2, SO2 and PO4 emissions of three heating systems of greenhouses.

The work and the conclusions are interesting, although the end result is quite predictable.

For your publication, I think it would be interesting to make some improvements to the article:

Major Flaws:

In the discussion section, I miss the comparison of the results obtained by the authors with other previous studies that have been cited in the review of the literatures, such as “Vadiee, A. and V. Martin, Energy analysis and thermoeconomic assessment of the closed greenhouse - The largest commercial solar building. Applied Energy, 2013. 102: p. 1256-1266 "or" Ha, T. and I.K. Lee, Kyeongseok Kwon, Hong, Sewoon, Computation and field experiment validation of greenhouse energy load using building energy simulation model. Int J Agric & Biol Eng, 2015. "

  • What are the differences between the results obtained by these studies and those obtained in this article?
  • What reasons do you think produce these differences?
  • are these differences congruent?
  • Are the results obtained in this article contradictory with those previously published by some other researcher?

CDV:  Thank you for your comment. To respond to your question, I have to clear the following:  First, I want to emphasize that the above-mentioned references focused only on the energy simulation of greenhouse buildings, not the environmental assessment. This means that the result of this current paper and the above-mentioned papers were relatively different in terms of results. For instance, the study of Vadiee et al (2013) and  Ha et al (2015) focused on the total energy load simulation of building structure using the energy simulation software whereas, in this current study, we focused on the total environmental burden of manufacturing and operating heating systems in greenhouse buildings. As illustrated in the research flow chart and mentioned in the manuscript, in this study, the final BES simulation result used in the paper was obtained from the study of Lee et al which is currently under review. The result of the initial part of this result calculated the energy load of the building structure and was used as an input value in the environmental assessment tool. Since the final result of this manuscript focussed on the environmental burden, it would be very difficult to compare the current research result and the above-mentioned references in the discussion. However, in the paper of Lee et al (under review), authors have discussed the comparison of simulation result and result of the literature and ensured that the issues mentioned in this comment were properly addressed in the manuscript.

Secondly, from the literature review (Chapter 2) of the manuscript, reference 42 to 45,  50 and more deals with environmental assessment for greenhouse crop production. However, it must be noted that the functional units used in these references were different. For instance, reference 43 has a functional unit of one kg of fresh (lettuce) or grain (barley) marketable product, reference # 50 has a functional unit is 1 hectare of cropped tomatoes over one cropping season. Reference 52 used the functional unit of one piece of Primrose while in our study, we used the 1 m2 of heated greenhouse per year. As implied in some LCA reviews (https://doi.org/10.1016/j.jclepro.2004.06.004), a comparison between assessment result is possible if we are comparing with alternative with similar functions (for example, different heating systems used for greenhouse). Moreover, according to Millet et al (2007) (https://doi.org/10.1016/j.jclepro.2005.07.016) the comparison of LCA result between different functional units, development stage, specification and others is unsuitable. In summary, we can not directly compare the assessment result cited in the literature and the result of the assessment in our study.

Using this information, in the discussion part, the only thing we could do is to directly compare the assessment result between the case scenario. This is the reason also why in section 4.3, we extended our comparison not only to the environmental burden, but also the economic and practical aspects. Moreover, from line 492 to 497, we compare the assessment result based on ranking (see the statement: “This is agreed by the conclusion stated by [37, 49, 58] which stated that a boiler has the highest carbon footprint compared to a different type of heat pumps systems”) and not based on quantitative values.

Minor Flaws:

  • It is not usual to include acronyms in the title, except in those cases in which they are widely known in the scientific field.

CDV: Thank you for the comment. The suggestion was well incorporated into the revised manuscript. Considering this opinion, the revised title of our manuscript is shown in the next line:

Integrated Building Energy Simulation-Life Cycle Assessment (BES-LCA) approach for environmental assessment of agricultural building: a review and application to greenhouse heating systems

  • Lines 67-68: this paragraph corresponds to references 7-9, while the next paragraph, lines 69-70 would correspond to reference 6.

CDV: I have thoroughly checked and updated the references according to the suggestions made by the reviewers. 

  • Lines 75-77. The authors state that “the qualitative amount of gas emission to the atmosphere of different heating and cooling system for crop production used in the greenhouse is also inadequate”. Why?

CDV: Thank you for pointing this out. I miswrote this sentence. What I mean by this sentence is, at present, there were only very few studies about the life cycle assessment for different heating and cooling systems in greenhouses. I revised the content like this:

Line 72-75: However, there were very few studies related to environmental impact assessment of heating and cooling systems in greenhouses since published paper usually relate heating and cooling systems to residential, commercial or industrial buildings and other applications ([10-13].

  • Line 475: “This is agreed by the conclusion stated by Toledo et al. (2019) and Samari et al., (2019) which stated that… ”Reference numbers must be included and the results obtained by these authors must also be included in section 2. Review of literatures.

CDV: Thank you for specifying this. I modified some part of the result and discussion. Especially, as can be seen in line 483 to 486 (previously line 475). I have fixed the citation of the references, added additional authors which have the same conclusion and have included the cited references in chapter 2 which is the Review of Literatures. Please refer to line 142 for example.

  • Line 165: “carbon dioxide CO2”. You have to eliminate CO2 because it is repetitive.

CDV: CO2 was deleted from the manuscript.

  • Line 193: correct “mongo” by “mango”

CDV: The mispelled word was changed according to the reviewer’s comment.

  • Line 217: reference 26 does not correspond to studies carried out in greenhouses, so it should be eliminated.

CDV: Thank you for the keen observation. The previous reference 26 was deleted from the manuscript (See line 227). All other references were also updated and changed according to the reviewers' comments and suggestions.

  • Line 267: correct “appropriate” by “appropiate”

CDV: I doubled checked the term. “Appropriate” was the right word for this sentence.

  • Line 321: correct “in the fan coal” by “in the fan coil”

CDV:  The term was misspelled. Fan coal was changed to “fan coil” as pointed out by the reviewer.

  • Line 360: You must include the reference and its number: "the study result conducted by Cecconet et al in 2020 for heat ..."

CDV: The reference was updated. For line 360, Cocconet et al  (2020) was assigned as reference number -

Round 2

Reviewer 2 Report

The justification provided by the authors in their cover letter about my comments regarding the discussion section are adequate from my point of view. Also, the modifications made in the discussion section on lines 501-502 seem adequate and I think they improve the understanding of the results.

Furthermore, all comments made as minor flaws have been conveniently corrected.
I consider that the article has improved in its writing and content, so I recommend its publication in Agronomy.